# Source and Transformation of MgO-Based Inclusions in Si-Mn-Killed Steel with Lime-Silicate Slag

**Jiaqi Zhao [1,2], Jianhua Chu [3], Xin Liu [1], Min Wang [1], Xiaofeng Cai [2], Han Ma [2] and Yanping Bao [1,***

1   State Key Lab of Advanced Metallurgy, University of Science & Technology Beijing, Beijing 100083, China
2   Institute of Research of Iron & Steel, Zhangjiagang 215600, China
3   School of Metallurgical Engineering, Anhui University of Technology, Maanshan 243002, China
*   Correspondence: baoyp@ustb.edu.cn

**Abstract:** The origin, evolution, and formation mechanism of MgO-based inclusions in Si-Mn-killed steel were studied in industrial trials with systematical samplings of the refining ladle, casting tundish, and as-cast bloom. In the present study, there were large numbers of MgO-based non-metallic inclusions, which started to form in the LF final process, and the MgO content in the lime-silicate slag increases from LF to VD process. The reason for the formation of MgO-based inclusions in refining process was analyzed using FactSage8.1 software. It was found that MgO-based inclusions were caused by the violent reaction between the slag and steel and the serious erosion of MgO-C refractory. The MgO solubility decreased in the lime-silicate slag and precipitated the periclase phase with basicity increasing. The solubility of MgO increased with an increase in the temperature. Measures were taken to optimize the refining process based on the above result. By increasing the slag basicity and increasing the content of MgO in the slag, erosion of the MgO-C refractory was reduced and the number of MgO-based non-metallic inclusions decreased from 0.2 to 0.04 per square millimeter.

**Keywords:** MgO saturation; industrial trial; non-metallic inclusions; thermodynamic calculation

## 1. Introduction

Inclusions which are higher in hardness and melting point are harmful to high-strength steel rods because inclusions often act as breakage initiation sites when subjected to cyclic stress. It would be much more desirable for the remained inclusions to be small in size and to demonstrate improved deformability, which is affected by the melting points of the inclusions themselves [1,2]. Therefore, inclusions, such as alumina or magnesia which are higher in melting point and hardness, should be avoided. For these reasons, Si-Mn complex deoxidation combined with low-basicity silicate slag refining is generally usedfor such steels [3–5].

The common type of non-metallic inclusions in these steels is CaO-SiO$_2$-Al$_2$O$_3$, usually together with some MgO [4]. Many researchers studied the effect of the Al$_2$O$_3$ and SiO$_2$ content on inclusions in Si-Mn-killed steel with lime-silicate slag, through either experiments or thermodynamic predictions [5–15]. Li Y et al. [5] studied the influence of a MgO refractory on inclusions and found thatinclusions containing MgO originated from the reduction of MgO in refractory or slag. Liu et al. [6] found that C in the MgO-C refractory also reduced MgO in the refractory and supplied Mg to the steel melt. Harada et al. [7] observed that the Al in the steel melt reduced the MgO in the refractory and Mg was dissolved into the steel melt. According to this mechanism, MgO in the MgO-C refractory supplies Mg to the steel melt owing to the reduction in Al in the steel. Chen L et al. [8] found that the refractory/steel interaction under vacuum conditions has a far more significant contribution to the increase of the [Al]$_S$ (acid-soluble Al) content in steel compared with the slag/steel interaction, which further leads to the generation of Al$_2$O$_3$-rich inclusions. Wang K et al. [9] studied the formation mechanism of CaO-containing inclusions in tire

cord steel. Park J et al. [10], Chen S et al. [11] and Liu N et al. [12] also studied the effect of refining slag basicity on non-metallic inclusions in steel. Additionally, Zhang L et al. [13–15] thought that kinetic factors were also of great importance in the formation of non-metallic inclusions. Moreover, many researchers [16–18] have studied the solubility of MgO in the CaO-SiO₂-FetO-MgO system. The solubility of MgO in low-basicity lime-silicate slag decreases with increasing the slag basicity (CaO/SiO₂). However, research and control on the silicate inclusions surrounding bulk MgO are still limited.

In the current study, industrial experiments were performed to investigate the composition and morphology of inclusions in the whole process including the steelmaking process. Based on the experiment results and thermodynamic calculation, the formation mechanism of MgO-based non-metallic inclusions is put forward.

## 2. Experiments

### 2.1. Experiment and Samplings

An industrial trial was carried out in a steel plant for Si-Mn-killed steel with the chemical composition listed in Table 1. The steelmaking process featured BOF (basic oxygen furnace steelmaking),-LF (ladle refining),-VD (vacuum degassing),-TD (soft blowing-tundish),-CC (continuous casting, 150 mm × 150 mm section). During tapping of the BOF, FeSi and Mn were used as deoxidizers and alloying elements. Besides, Cr–Fe alloy and refining slag material were also added. In the LF process, Si–Fe and Mn were employed for composition modification. Soft blowing was couducted for about 25 min to remove inclusions after VD degassing. Finally, the molten steel was sent to continuous casting.

**Table 1.** Chemical composition of the experimental steel, wt%.

| C | Si | Mn | Cr | P | S | Als |
|---|---|---|---|---|---|---|
| 0.50–0.55 | 0.30–0.35 | 0.40–0.50 | 0.20–0.30 | <0.014 | <0.012 | <0.0035 |

To investigate the source and transformation of MgO-based inclusions, steel samples and slag samples were taken with samplers (diameter of 30 mm, thickness of 10 mm): three from LF refining (start, after alloying, end), two from VD refining (start, end), one from soft blowing, one from the tundish, and one from billet casting, as shown in Figure 1.

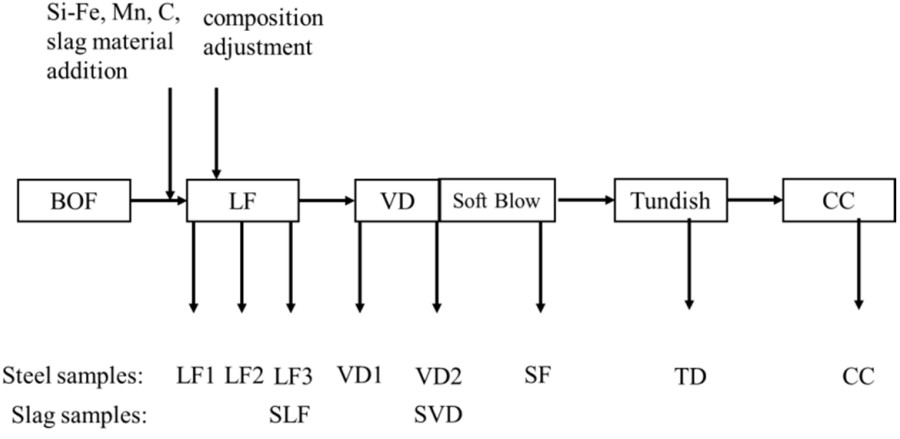

**Figure 1.** Process of industrial trials and sampling locations.

### 2.2. Analysis of Samples

These samples were polished for characterization of the inclusions using scanning electron microscopy (SEM). The composition of inclusions was analyzed by an energy dispersive X-ray spectrometer (EDS) attached to the SEM. The slag composition shown in Table 2 was determined using an X-ray fluorescence spectrometer (XRF) with a 5 pct relative standard deviation.

**Table 2.** Chemical composition of the refining slag, wt%.

| Sampling Location | CaO | SiO$_2$ | Al$_2$O$_3$ | MgO | T.Fe+MnO |
|---|---|---|---|---|---|
| SLF | 39–44 | 37–43 | 5–10 | 3–7 | ≤4 |
| SVD | 38–42 | 37–45 | 4–8 | 8–13 | ≤5 |

## 3. Results and Discussion

### 3.1. Evolution of Non-Metallic Inclusions in Steelmaking Process

Inclusions were analyzed using an SEM equipped with an EDS in each metallographic sample taken from different stages, as shown in Figure 2. The types of non-metallic inclusions wre CaO-Al$_2$O$_3$-SiO$_2$ and Al$_2$O$_3$-SiO$_2$-MnO at the early stage of the LF process, but CaO-Al$_2$O$_3$-SiO$_2$-MgO was found in the VD process. During the later stage of the LF process, there were more MgO inclusions, which were irregular and wrapped by CaO-Al$_2$O$_3$-SiO$_2$ inclusions. Those inclusions were not completely removed in the process of soft stirring and tundish pouring, and they eventually remained in the billet. The variation in the average mass fraction of MgO and Al$_2$O$_3$ in inclusions during the refining process is shown in Figure 3. The average Al$_2$O$_3$ content of oxide inclusions gradually decrease from 43% to 15% in the refining process. The average MgO content of oxide inclusions increase from 7% to 15%. The average MgO content in inclusions increased rapidly from the late refining stage to the VD stage.

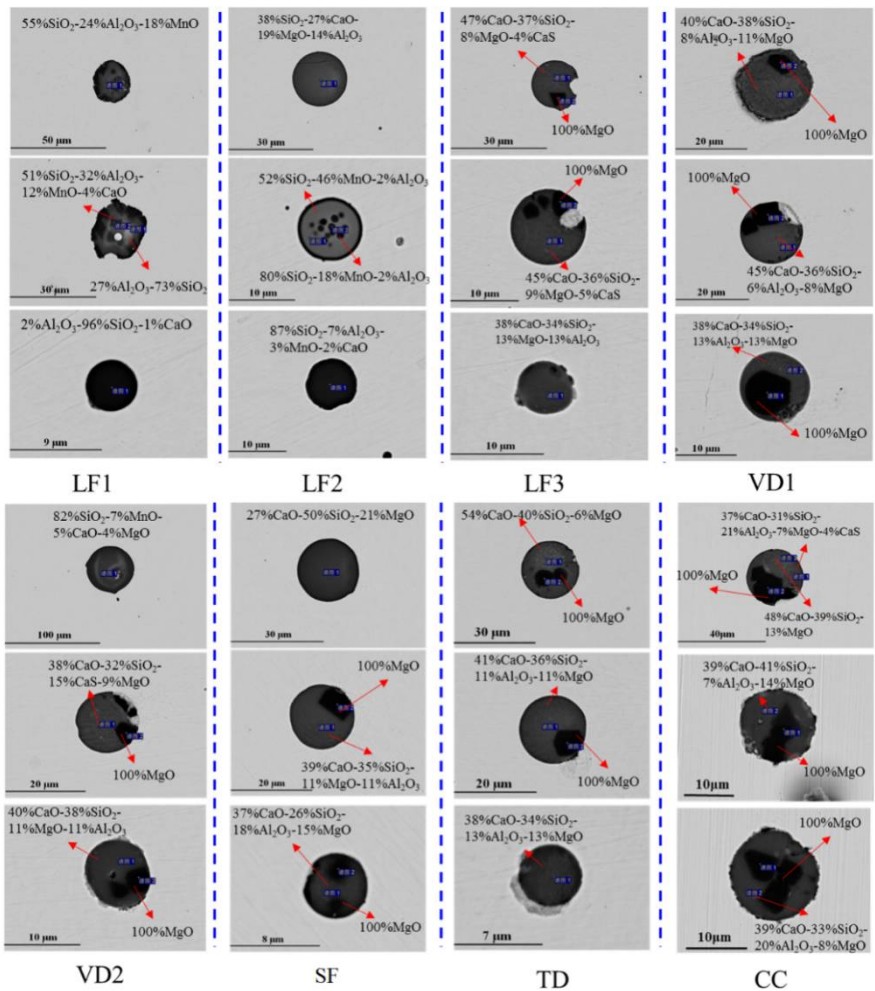

**Figure 2.** Morphology and composition of typical inclusions in steel.

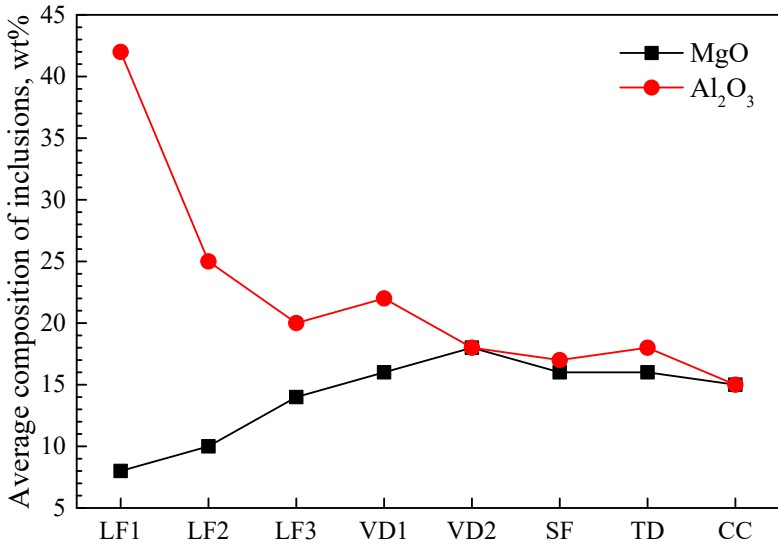

**Figure 3.** Average mass fraction of $Al_2O_3$ and MgO in $CaO$-$Al_2O_3$-$MgO$-$SiO_2$ inclusions.

### 3.2. Analysis of the MgO-C Refractory after Continuous Casting

By taking samples of refractory materials at the slag line of the ladle furnace after smelting, the longitudinal section of the refractory contacting the molten steel was analyzed with a scanning electron microscope, as shown in Figure 4. From inside to outside, it can be divided into three layers, including the complete reaction layer, transition layer and matrix. Almost all C in the reaction layer and transition layer of the refractory was oxidized, and the reaction layer infiltrated obvious Ca-Si-Al-O inclusions and Fe droplets. Further analysis of the micro area in the reaction layer is shown in Figure 5. There was a large amount of massive MgO in the refractory, which was wrapped by infiltrated Ca-Si-Al-O inclusions.

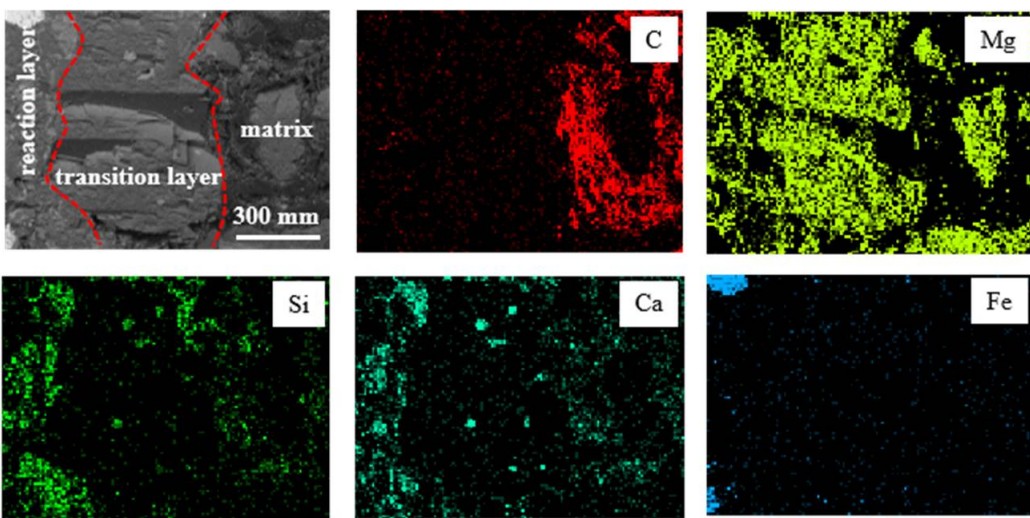

**Figure 4.** Longitudinal sectional microstructure and element mapping of the ladle furnace slag line MgO-C bricks.

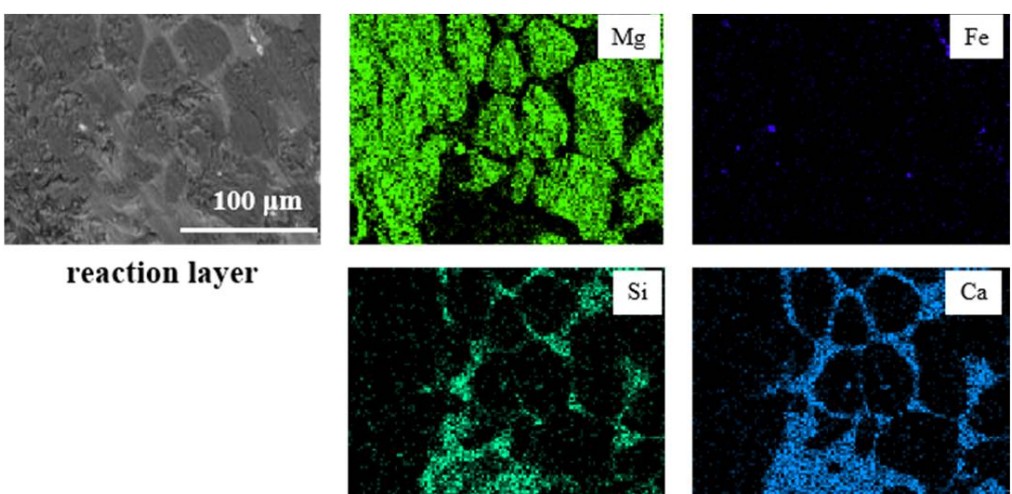

**Figure 5.** Microstructure and element mapping of the reaction layer in the slag line MgO-C bricks.

### 3.3. Source of MgO-Based Non-Metallic Inclusions

Inclusions were analyzed using an SEM equipped with an EDS in the billet as shown in Figure 6. It was found that the inclusions in steel were $CaO-Al_2O_3-SiO_2$ and $Al_2O_3$-$SiO_2$-MnO, and that there were a large number of composite inclusions with massive MgO wrapped by silicate inclusions. The number of MgO-based inclusions in the billet is shown in Table 3. The size distribution of all inclusions and MgO-based inclusions in billet is shown in Figure 7, and the morphology and composition of typical inclusions are shown in Figure 8. It can be seen that the light gray part of the inclusion corresponds to the $CaO-SiO_2-Al_2O_3$ phase, and the black area depicts the MgO phase. The shape of the MgO phase is irregular and has edges and corners.

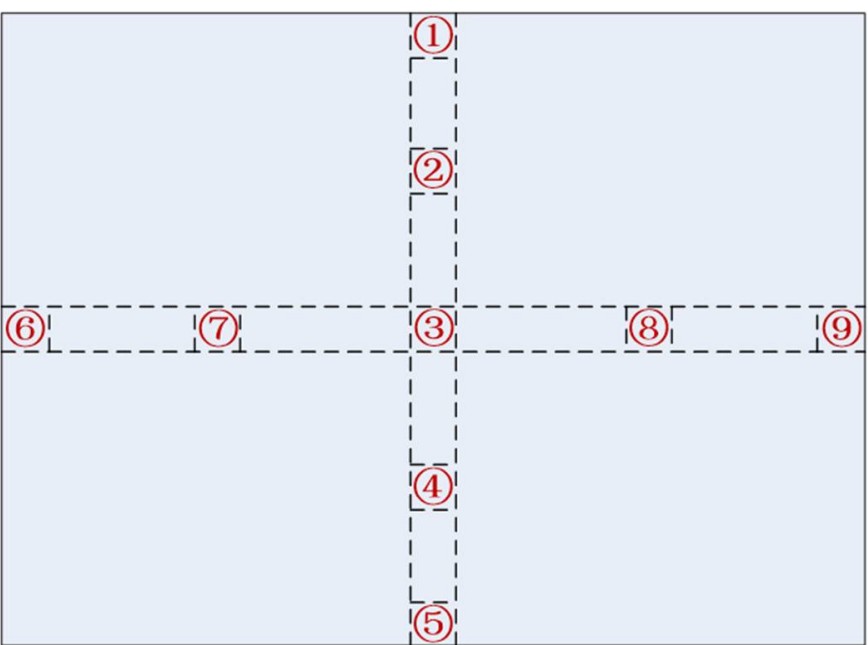

**Figure 6.** Sampling plan.

**Table 3.** Non-metallic inclusions in billet before improvement.

| NO. | Number of All Types of Inclusions | MgO-Based Non-Metallic Inclusions | |
|---|---|---|---|
| | | Number | Density, /mm² |
| 1 | 177 | 28 | 0.32 |
| 2 | 204 | 26 | 0.29 |
| 3 | 272 | 5 | 0.06 |
| 4 | 148 | 14 | 0.16 |
| 5 | 134 | 9 | 0.10 |
| 6 | 333 | 15 | 0.17 |
| 7 | 298 | 22 | 0.25 |
| 8 | 332 | 23 | 0.26 |
| 9 | 207 | 15 | 0.17 |
| Average | | | 0.20 |

The MgO inclusions probably came from the refining slag or corrosion of the MgO-C refractory. Refining slag with a low basicity has strong corrosiveness. The MgO content in the refining slag increased from the LF to the VD process, as shown in Table 2. The thermodynamic FactSage8.1 software with FToxid databases, SafeNet Inc., Belcamp, MD, USA (for oxides) was used to simulate the precipitation of inclusions during solidification. Combined with the statistical data of the inclusions mentioned above, the average composition of inclusions in the steel was $40CaO-30SiO_2-15Al_2O_3-15MgO$ in the billet. At the molten steel temperature, there are precipitations of MgO from liquid inclusions in the $40CaO-30SiO_2-15Al_2O_3-15MgO$ system, as shown in Figure 9.

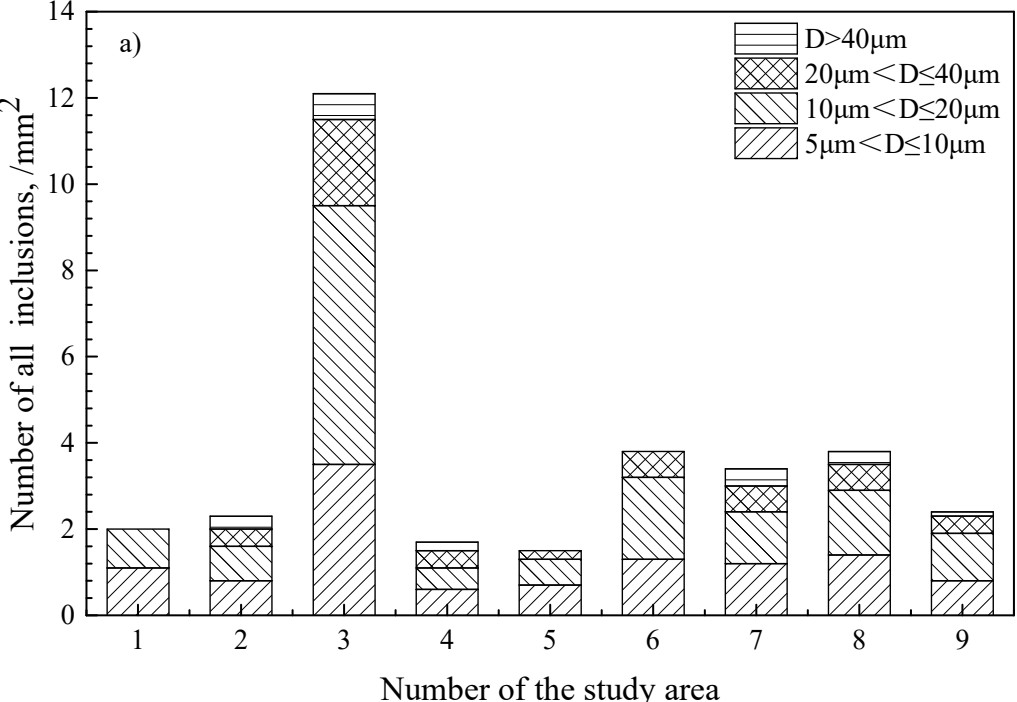

**Figure 7.** *Cont.*

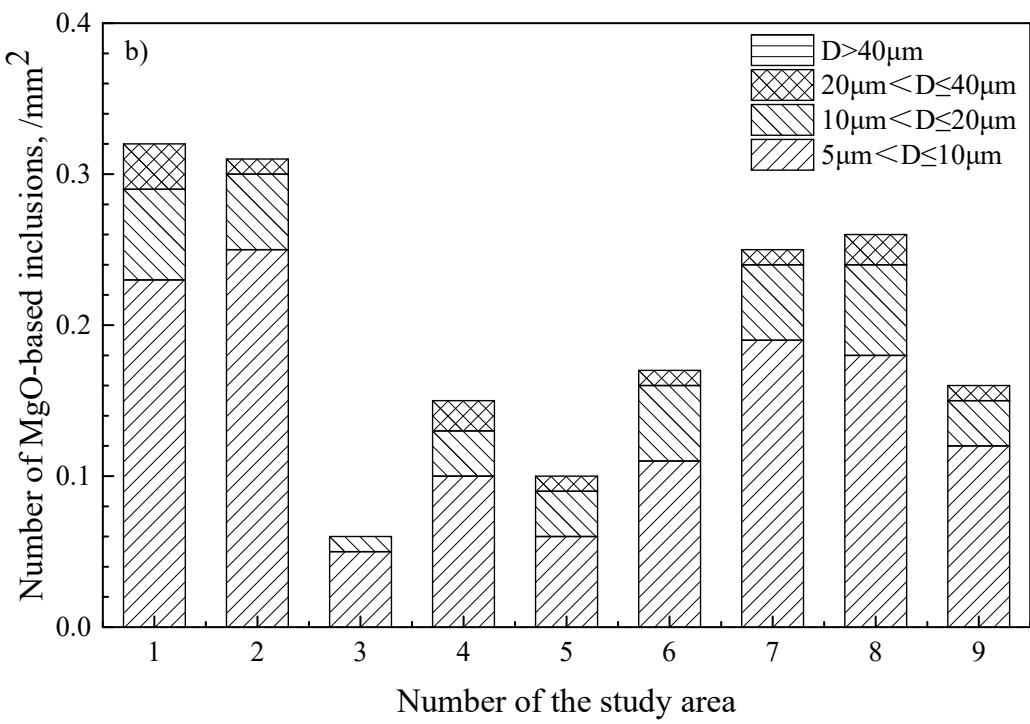

**Figure 7.** Size distribution of inclusions in the billet. (**a**) all types of inclusions; (**b**) MgO-based inclusions.

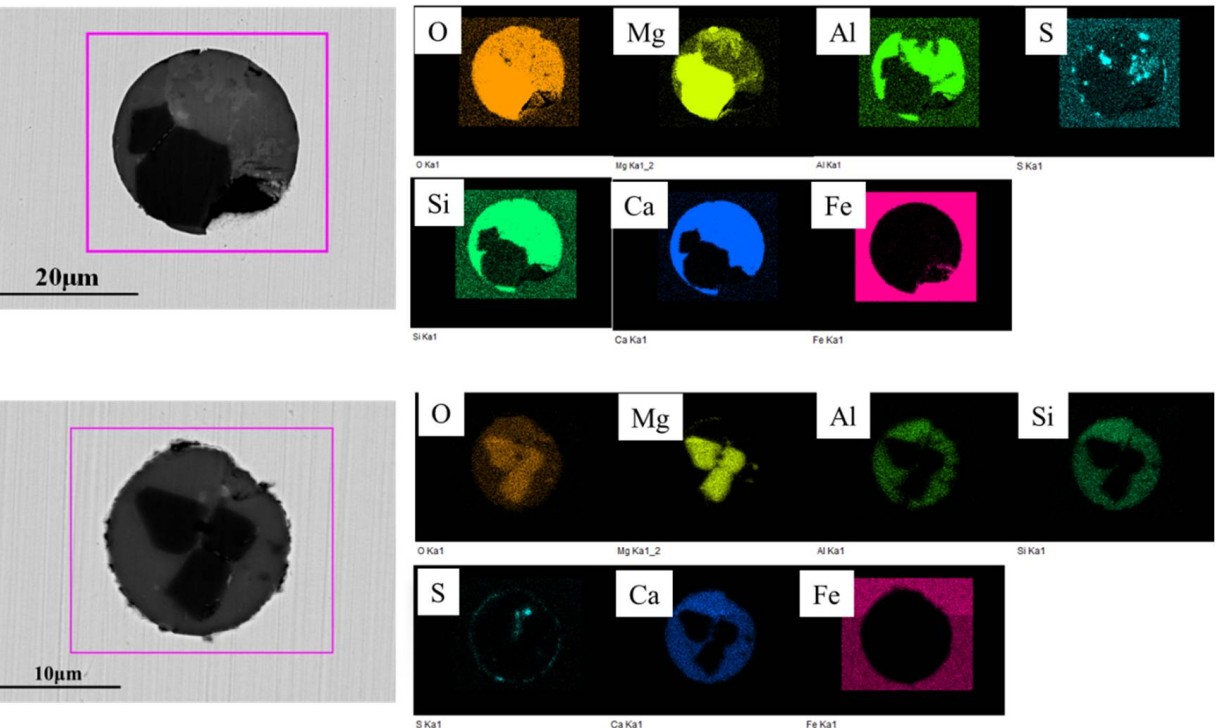

**Figure 8.** Morphology and composition of typical inclusions in billet.

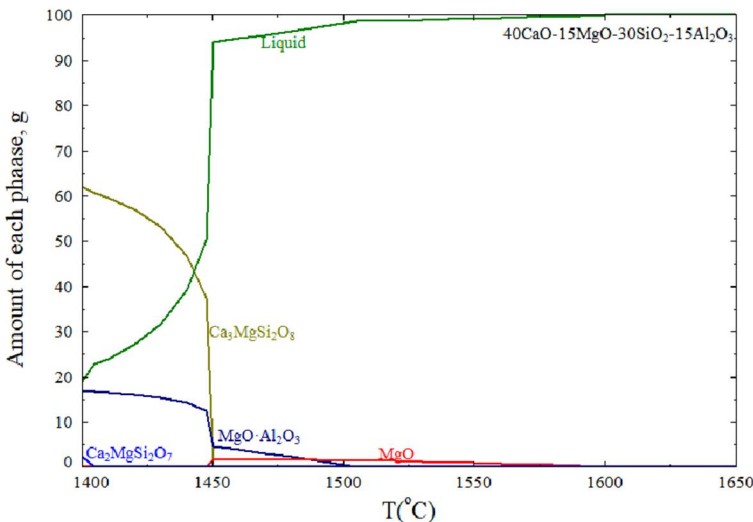

**Figure 9.** Precipitation of inclusions during solidification.

### 3.4. Thermodynamic Considerations of MgO Saturation in Slag

MgO-unsaturated secondary metallurgical slag can cause MgO to dissolve from a refractory brick into slag. The MgO solubility in slag is controlled by saturation phases that include periclase (MgO), spinel (MgO·Al$_2$O$_3$) or forsterite (2MgO·SiO$_2$) in the quaternary CaO-SiO$_2$-Al$_2$O$_3$-MgO system. The thermodynamic FactSage8.1 software was used to obtain the MgO saturation concentration under different conditions by the isothermal stability diagrams (ISDs) [19].

Figure 10 shows the effect of the Al$_2$O$_3$ content on MgO solubility with 25~40% SiO$_2$ contents at 1873 K. The MgO solubility was about 2~34% with 0~10% Al$_2$O$_3$ content in the quaternary CaO-SiO$_2$-Al$_2$O$_3$-MgO system. With an increase in the Al$_2$O$_3$ content, the MgO solubility in the MgO-saturated phases periclase (MgO), forsterite (Mg$_2$SiO$_4$) and dual saturated (MgO and Ca$_2$SiO$_4$) increased, and thereafter, decreased in the MgO-saturated phase spinel (MgO·Al$_2$O$_3$). Additionally, the SiO$_2$ content had a significant effect on the MgO solubility in the lime-silicate slag. The MgO solubility increased with an increase to less than 40% SiO$_2$ content.

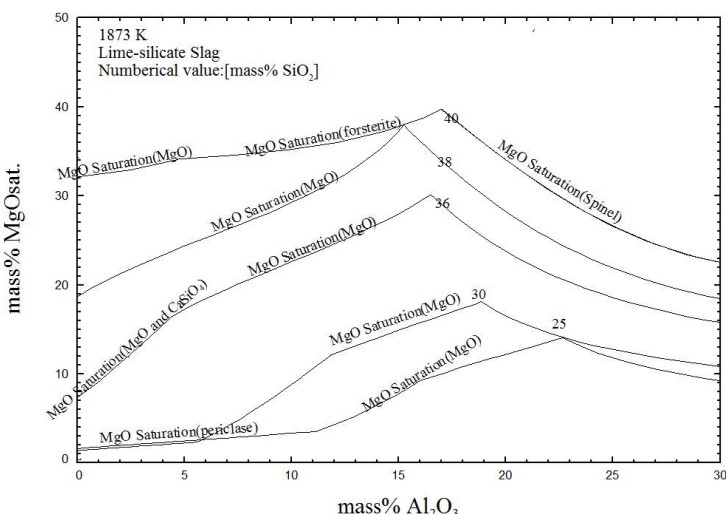

**Figure 10.** Effect of the Al$_2$O$_3$ content on MgO solubility in lime-silicate slag at 1873 K.

Figure 11 displays the effect of the Al$_2$O$_3$ content at different basicities (CaO/SiO$_2$) on MgO solubility at 1873 K. It is notable that slags with higher basicity were saturated with less MgO. Typical lime-silicate slag had less than 10% Al$_2$O$_3$ content and 0.8~2.0 basicity,

therefore, the solubility of MgO was 1~29%. This conclusion was consistent with previous researches [7–9] considering the solubility of MgO in the $CaO-SiO_2-Fe_tO-MgO$ system.

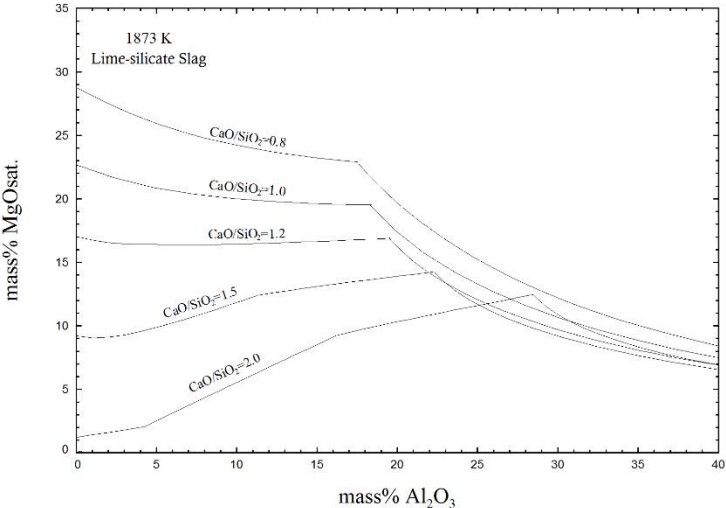

**Figure 11.** Effect of basicity ($CaO/SiO_2$) on MgO solubility with a low basicity of the slag at 1873 K.

Figure 12 shows the effect of the $Al_2O_3$ content at different temperature on the MgO solubility with 40% $SiO_2$ content. More MgO was saturated in the quaternary $CaO-SiO_2-Al_2O_3-MgO$ system at a higher temperature.

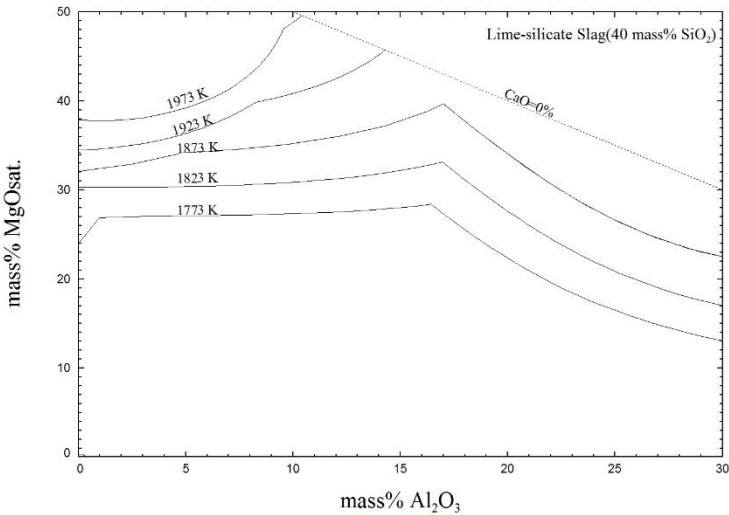

**Figure 12.** Effect of the $Al_2O_3$ content on MgO solubility in slag with 40% $SiO_2$ at different temperatures.

*3.5. Evolution Mechanism of MgO-Based Non-Metallic Inclusions*

The main components of refractory materials are MgO and C. Because the MgO content in the slag is not saturated, the basicity of the refining slag is low, and the oxidation property is high, the C in the refractory is oxidized, and MgO gradually enters into the slag, resulting in the gradual increase of the MgO content in the slag and the gradual erosion of the refractory [20]. At the later stage of smelting, the impurities in the steel are mainly $CaO-SiO_2-Al_2O_3$ low melting point liquid inclusions, the C in the refractory is oxidized, and the CaO-SiO2-$Al_2O_3$ inclusions in the steel enter the surface cracks of the refractory. As Liu [6] found, C in the MgO-C refractory also reduced MgO in the refractory in Al-killed steel. The main difference was that a MgO-based inclusion was not formed by Mg from the refractory in the steel. With the scouring of the refractory by the bottom blowing of the ladle and the flow of the steel, the massive MgO flakes into the steel, or the CaO-SiO2-Al2O3

wraps the massive MgO inclusions and washes into the steel. The detailed mechanism process is illustrated in Figure 13. The evolution mechanism of MgO-based inclusions is shown in Figure 14.

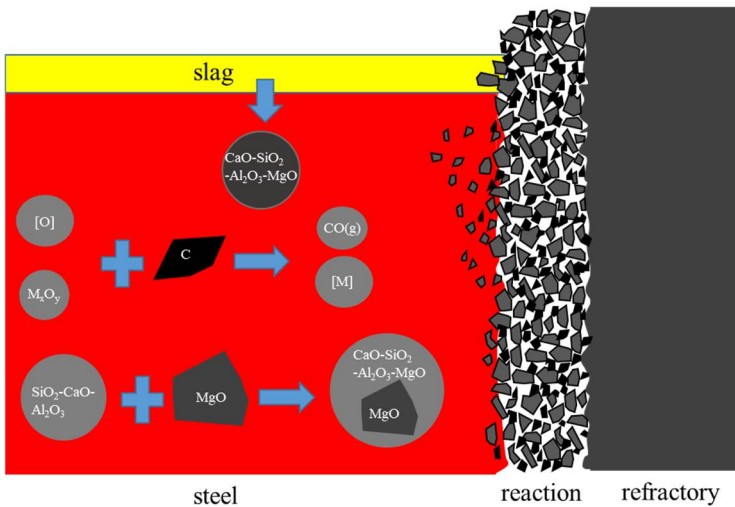

**Figure 13.** Illustration diagram of steel-slag-refractory interaction mechanism.

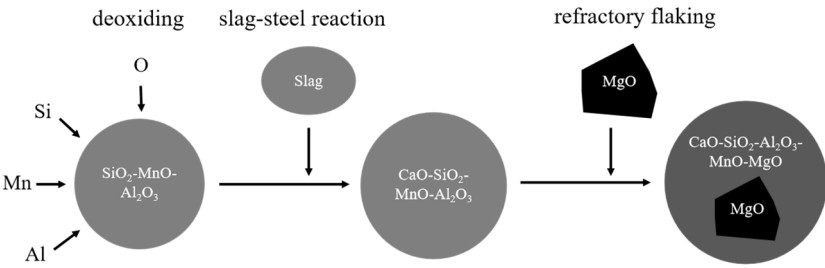

**Figure 14.** Schematic diagram of MgO-based inclusion evolution mechanism.

### 3.6. Improvements in Industrial Trial

From the analysis above, the corrosion of the refractory can be reduced by increasing the basicity and MgO content in the refining slag. Hence improvements were made in the industrial trial. We redesigned the refining slag, specifically by: (1) appropriately increasing the amount of lime in the LF refining process, increasing the amount of lime by 50–100kg, controlling the basicity of slag to 1.5–2.0 and reducing the saturation solubility of MgO in refining slag; (2) increasing the content of MgO in the refining slag, making the content of MgO in the refining slag reach saturation state and reducing the corrosion of the refractory. The main components of the refining slag before and after improvement are shown in Table 4.

**Table 4.** Components of the refining slag in LF before and after improvement, wt%.

| Different Process | CaO | $SiO_2$ | $Al_2O_3$ | MgO | T.Fe+MnO |
|---|---|---|---|---|---|
| before improvement | 39–44 | 37–43 | 5–10 | 3–7 | ≤5.0 |
| after improvement | 45–52 | 27–33 | 3–8 | 8–14 | ≤1.5 |

Inclusions were analyzed using an SEM equipped with an EDS in the billet, as shown in Figure 15. After improvement, the samples of the continuous casting billet were analyzed. The number and density of inclusions are shown in Table 5. The morphology and composition of typical inclusions in billet are shown in Figure 16. The inclusions in the steel were $CaO$-$Al_2O_3$-$SiO_2$ and $Al_2O_3$-$SiO_2$-$MnO$, some of which also contained a minor

amount of MgO, but the composition of MgO was relatively uniform in distribution, and the number of composite inclusions wrapped with MgO decreased greatly.

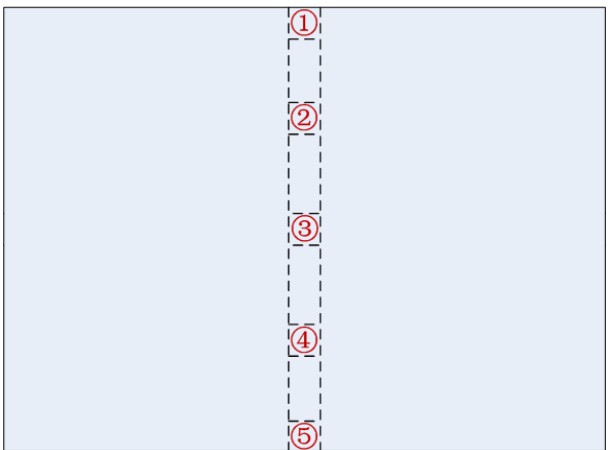

**Figure 15.** Sampling plan.

**Table 5.** Non-metallic inclusions in the billet after improvement.

| NO. | Number of all Types of Inclusions | MgO-Based Non-Metallic Inclusions | |
|-----|-----------------------------------|--------|------------------|
| | | Number | Density, /mm$^2$ |
| 1 | 168 | 2 | 0.03 |
| 2 | 243 | 5 | 0.06 |
| 3 | 261 | 5 | 0.06 |
| 4 | 177 | 2 | 0.02 |
| 5 | 244 | 4 | 0.05 |
| Average | | | 0.04 |

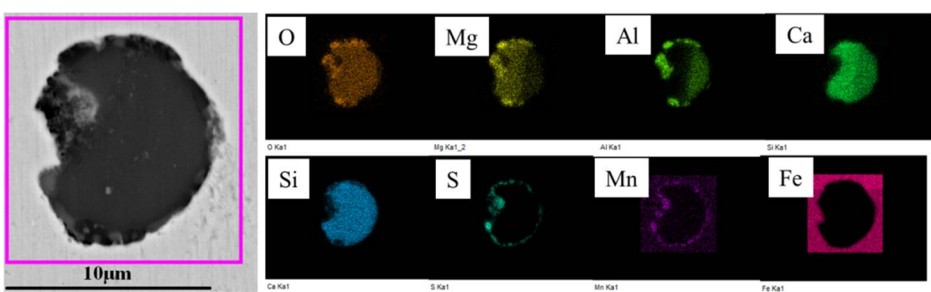

**Figure 16.** Morphology and composition of typical inclusions in billet.

Before improvement the average density of oxide inclusions containing MgO was 0.59/mm$^2$, while that of massive MgO inclusions was 0.2/mm$^2$. After improvement, the average density of oxide inclusions containing MgO was 0.45/mm$^2$, and the density of MgO-based oxide inclusions was reduced to 0.04/mm$^2$ in Figure 17. The proportion of MgO-based oxide inclusions decreased from 6.8% to 1.7%. The objective of controlling MgO-based oxide inclusions was achieved by optimizing the refining process.

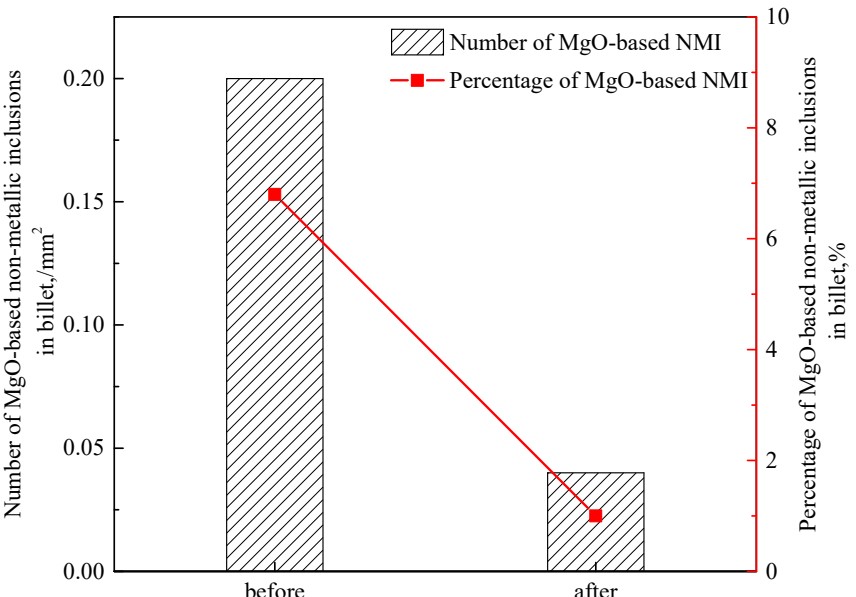

**Figure 17.** The number of MgO-based oxide inclusions in the billet.

## 4. Conclusions

In the present study, industrial trials and thermodynamic calculations were performed to investigate the solubility of MgO in the quaternary $CaO-SiO_2-Al_2O_3-MgO$ system in secondary metallurgical slag. The following conclusions were obtained:

1. By using an SEM equipped with an EDS, it was found that there were a large number of MgO-based non-metallic inclusions, which started to form in the LF final process. The content of MgO in the lime-silicate slag increased from the LF process to VD process, which may have been caused by the erosion of MgO-C refractory.
2. The solubility of MgO in low basicity lime-silicate slag decreased with an increase in the slag basicity ($CaO/SiO_2$). In the lime-silicate slag, the solubility of MgO increases with an increase in the temperature. The solubility of MgO was 1~29% in typical lime-silicate slag with 0~10% $Al_2O_3$ and 0.8~2.0 slag basicity ($CaO/SiO_2$) at 1873 K.
3. With the oxidation of C in the refractory and the scouring of the refractory by the bottom blowing of the ladle and the flow of moten steel, the massive MgO flakes into the steel, or the $CaO-SiO_2-Al_2O_3$ wraps the massive MgO inclusions and washes into the steel.
4. By increasing the slag basicity and increasing the content of MgO, erosion of MgO-C refractory was reduced and number of MgO-based non-metallic inclusions was decreased from 0.2 to 0.04 per square millimeter. The objective of controlling MgO-based oxide inclusions was achieved by optimizing the refining process.

**Author Contributions:** Conceptualization, J.Z. and J.C.; methodology, J.Z.; software, X.C.; validation, J.Z., J.C. and M.W.; formal analysis, J.C.; investigation, J.Z.; resources, J.Z.; data curation, J.Z.; writing—original draft preparation, J.Z.; writing—review and editing, X.L.; visualization, X.C; supervision, M.W., H.M. and Y.B.; project administration, H.M.; funding acquisition, J.Z. All authors have read and agreed to the published version of the manuscript.

**Funding:** This work was supported by the National Natural Science Foundation of China (Grant No. 51774031).

**Data Availability Statement:** The data presented in this study are available on request from the corresponding author.

**Conflicts of Interest:** No potential conflict of interest was reported by the authors.

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
