# Peer review of "Source and Transformation of MgO-Based Inclusions in Si-Mn-Killed Steel with Lime-Silicate Slag"

_metals, doi:10.3390/met12081323_

Round 1

Reviewer 1 Report

The work investigated the MgO inclusions in Si-Mn killed steel production, which may interest the readers and in the scope of our journal. However, the manuscript needs to be revised before further consideration. 

1), line 34, “Al2O3 and SiO2…”  subscript,

2), “mass pct. and wt%”, please keep the units in the manuscript be consistent,

3), line 79, “Figure 2. Morphology and composition of typical inclusions in steel” should be better. Why use point data instead of map scanning data?

4) Table 3 & Table 5, how about the particle size distribution of these inclusions?

5), Figure 4b, the sum of the amount of the phase is not equal to 100 when the temperature is less than 1450C. Please check your calculation.

6), Figure 8, What is the composition of magnesia carbon brick? What are the black ones in refractory, MgO or C? The main body of the refractory should be MgO, so I think the grey clour presents the MgO, but the black in the inclusions is MgO, which is very confusing.

7), Corrosion of the refractories occurs due to the chemical attack while erosion involves both the chemical attack (erosion) and the mechanical abrasion of the refractory. According to the shape of the MgO in the inclusions, more like the result of a physical abrasion. What do the authors think?

8), the slag properties are also important for the process. What are the changes in the physical and chemical properties of the slag after modification, and what are the effects on inclusions?

9), Figure 10, a double Y figure should be better for presenting.

Author Response

Responses to Comments

Dear editor and Reviewers,

On behalf of my co-authors, we thank you very much for giving us an opportunity to revise our manuscript, and we appreciate you and reviewers very much for your positive and constructive comments and suggestions on our manuscript entitled “Source and Transformation of MgO-Based Inclusions in Si-Mn Killed Steel with Lime-Silicate Slag”. (Manuscript ID: metals-1783655). We have studied reviewer’s comments carefully and have made revision which marked in red in the paper. Meanwhile, we have responded point by point to all the comments as listed below. Attached please find the revised version, which we would like to submit for your kind consideration.

To Reviewer 1

Comment: The work investigated the MgO inclusions in Si-Mn killed steel production, which may interest the readers and in the scope of our journal. However, the manuscript needs to be revised before further consideration.

Reply: We are thankful to the reviewer for the kind recommendation and careful review.

(1) Comment:  line 34, “Al2O3 and SiO2…”  subscript,.

Reply: Thanks for your valuable comment. The subscript has been corrected in the revised manuscript.

(2) Comment: mass pct. and wt%”, please keep the units in the manuscript be consistent.

Reply: Thanks for your kind suggestion. The units in the manuscript have been consistent in the revised manuscript.

(3) Comment: line 79, “Figure 2. Morphology and composition of typical inclusions in steel” should be better. Why use point data instead of map scanning data?

Reply: Thanks for your kind suggestion. The title of Figure 2 has been replaced. The map scanning data is indeed more accurate than the point scanning data, and the heterogeneity in some inclusions is small, so the map scanning data cannot be used. In addition, the actual scanning of the point scanning of the electron microscope also includes the map around the point.

(4) Comment: Table 3 & Table 5, how about the particle size distribution of these inclusions?

Reply: Thanks for your valuable comment. The particle size distribution of these inclusions in Fig. 5 has been added in the revised manuscript.

(5) Comment: Figure 4b, the sum of the amount of the phase is not equal to 100 when the temperature is less than 1450C. Please check your calculation.

Reply: Thanks for your kind suggestion. Figure 4b has been recalculated in the revised manuscript, please see the revised Figure 7.

(6) Comment: Figure 8, What is the composition of magnesia carbon brick? What are the black ones in refractory, MgO or C? The main body of the refractory should be MgO, so I think the grey clour presents the MgO, but the black in the inclusions is MgO, which is very confusing.

Reply: Thanks for your kind suggestion. Figure 8 has been corrected in the revised manuscript, please see the revised Figure 11.

(7) Comment: Corrosion of the refractories occurs due to the chemical attack while erosion involves both the chemical attack (erosion) and the mechanical abrasion of the refractory. According to the shape of the MgO in the inclusions, more like the result of a physical abrasion. What do the authors think?

Reply: Thanks for your valuable comments. The author thinks that the corrosion of refractory materials includes not only the reaction between refining slag and refractory materials, but also the physical abrasion or physical scouring of molten steel or slag on refractory materials.

(8) Comment: the slag properties are also important for the process. What are the changes in the physical and chemical properties of the slag after modification, and what are the effects on inclusions?

Reply: Thanks for your valuable comments. After modification, the content of MgO in refining slag and the basicity of refining slag were increased. The inclusions in steel are CaO-Al2O3-SiO2 and Al2O3-SiO2-MnO, some of which contained a little amount of MgO, but the composition of MgO was relatively uniform distribution, and the number of composite inclusions wrapped with MgO decreased greatly.

(9) Comment: Figure 10, a double Y figure should be better for presenting.

Reply: Thanks for your valuable comments. Figure 10 has been corrected in the revised manuscript, please see the revised Figure 14.

We have improved the manuscript in the manuscript. These changes will not influence the content and framework of the paper. And here we did not list the changes but marked in red in revised manuscript. We would like to express our great appreciation to you and reviewers for comments on our paper. Looking forward to hearing from you.

Thank you and best regards.

Sincerely yours,

Prof. Yanping Bao

State Key Laboratory of Advanced Metallurgy

University of Science and Technology Beijing

Beijing 100083, China

Jul. 8, 2022

Reviewer 2 Report

REVIEW COMMENTS

Manuscript ID: metals-1783655

Source and Transformation of MgO-Based Inclusions in Si-Mn Killed Steel with Lime-Silicate Slag

An interesting work. The authors analyze the formation of MgO-based inclusions at different stages of the steelmaking process through industrial trials. Using thermodynamic calculations, they analyze the influence of the basicity of the slag on the solubility of MgO. They obtain the conclusion that for low values of basicity, an erosion of the refractory can occur. As a consequence of the erosion of the refractory, contributing MgO particles to the molten steel. The authors propose an improvement of the refining process by increasing the basicity of the slag by adding lime. The tests carried out with this improvement have resulted in a lower number of MgO-based inclusions.

The work is interesting but some aspects I think need to be improved. I would appreciate it if you would consider the following suggestions:

Abstract

Lines 11-21: The abstract has numerous English language errors and is difficult to understand. For example: “The reason for the formation of MgO based inclusions in refining process is the violent reaction between slag and steel and the serious erosion of MgO-C refractory by using FactSage8.1 software.” I think the authors want to explain that “the reason for the formation of MgO… HAS BEEN ANALYZED by using FactSage8.1 software” or something like that. Please review the text completely.

Introduction

Lines 36-38: The authors state that there is a lack of systematic research on the origin and formation mechanism of MgO-based non-metallic inclusions in Si-Mn quenched steel. However, numerous works can be found in the scientific literature where they analyze the formation of inclusions with MgO in their composition. The authors must make a more detailed state of the art of the topic and clearly explain the contributions of their work with respect to others available in the literature.

Experiments

Line 58: Why were three samples taken in the Ladle Refining process? What is the difference between those three samples? Different times after the addition of the deoxidizers? I believe that the manuscript does not explain what the taking of these three samples contributes. Explain it more clearly, please.

Line 58: The same for the two samples taken in the Vacuum Degassing process.

Results and discussion

Lines 71-77: Has a quantitative analysis of MgO-based inclusions been carried out at each stage of the steelmaking process? I think it would be interesting to define the stage in which the formation of a greater number of this type of inclusions occurs.

Line 85. Table 3: I don't understand the information in the table. How is density calculated? What does the first column mean? Different samples analyzed? Does "Total" refer to the number of inclusions observed in each sample? Explain the information in the table more clearly, please.

Lines 86-88: Please describe in the text the characteristics (morphology and composition) of typical billet inclusions.

Lines 133-138: The formation mechanism of the MgO-based inclusions is consistent with the results of the thermodynamic analysis. If correct, I think that a lower number of MgO-based inclusions should be observed in the samples at the beginning of the Ladle Refining process. Do the authors have any evidence that this relationship?

Line 153: “but the composition was relatively uniform”. What do the authors mean by this sentece? Explain it more clearly.

Line 157. Table 5: Same comments as those made in table 3..

Lines 158-159: Please describe in the text the characteristics (morphology and composition) of typical billet inclusions. Compare the characteristics of the inclusions in Figure 9 and Figure 3.

Author Response

Responses to Comments

Dear editor and Reviewers,

On behalf of my co-authors, we thank you very much for giving us an opportunity to revise our manuscript, and we appreciate you and reviewers very much for your positive and constructive comments and suggestions on our manuscript entitled “Source and Transformation of MgO-Based Inclusions in Si-Mn Killed Steel with Lime-Silicate Slag”. (Manuscript ID: metals-1783655). We have studied reviewer’s comments carefully and have made revision which marked in red in the paper. Meanwhile, we have responded point by point to all the comments as listed below. Attached please find the revised version, which we would like to submit for your kind consideration.

To Reviewer 2

Comment: An interesting work. The authors analyze the formation of MgO-based inclusions at different stages of the steelmaking process through industrial trials. Using thermodynamic calculations, they analyze the influence of the basicity of the slag on the solubility of MgO. They obtain the conclusion that for low values of basicity, an erosion of the refractory can occur. As a consequence of the erosion of the refractory, contributing MgO particles to the molten steel. The authors propose an improvement of the refining process by increasing the basicity of the slag by adding lime. The tests carried out with this improvement have resulted in a lower number of MgO-based inclusions.

The work is interesting but some aspects I think need to be improved. I would appreciate it if you would consider the following suggestions:

Reply: We are thankful to the reviewer for the kind recommendation and careful review.

(1) Comment: Lines 11-21: The abstract has numerous English language errors and is difficult to understand. For example: “The reason for the formation of MgO based inclusions in refining process is the violent reaction between slag and steel and the serious erosion of MgO-C refractory by using FactSage8.1 software.” I think the authors want to explain that “the reason for the formation of MgO… HAS BEEN ANALYZED by using FactSage8.1 software” or something like that. Please review the text completely.

Reply: Thanks for your valuable comment. We have carefully gone over the full manuscript and corrected the English grammar errors in the revised manuscript.

(2) Comment: Lines 36-38: The authors state that there is a lack of systematic research on the origin and formation mechanism of MgO-based non-metallic inclusions in Si-Mn quenched steel. However, numerous works can be found in the scientific literature where they analyze the formation of inclusions with MgO in their composition. The authors must make a more detailed state of the art of the topic and clearly explain the contributions of their work with respect to others available in the literature.

Reply: Thanks for your valuable comments. There are many related reports on the inclusions containing MgO components, but there are few reports on the silicate inclusion surrounding bulk MgO as studied in this paper. This sentence in this article is not accurate enough and has been corrected in the revised manuscript.

(3) Comment: Line 58: Why were three samples taken in the Ladle Refining process? What is the difference between those three samples? Different times after the addition of the deoxidizers? I believe that the manuscript does not explain what the taking of these three samples contributes. Explain it more clearly, please.

Reply: Thanks for your valuable comments. In order to study the composition and morphology of the initial inclusions produced after deoxidation and alloying of converter tapping and slagging, the molten steel was mixed violently during converter tapping, and the inclusions were mainly the initial deoxidation products and some slag-steel reaction products; The purpose of sampling at LF mid-stage is to study the composition and morphology of inclusions after adding lime, fluorite and alloy after LF arrival. Because of adjusting the composition of slag and alloy element, there is a violent reaction between slag and steel by uniform bottom blowing and stirring in order to make the composition uniform, which could cause the transformation of the composition and type of inclusions; After the composition of molten steel and slag reaches the standard, and the temperature reaches the standard by continuing to power on, the bottom blowing is gradually reduced, and tapping is ready according to the production rhythm. The slag-steel reaction in this process is weak, but the time is relatively long. Sampling at the LF final-stage is to study the changes of inclusion composition and type in this process.

(4) Comment:  The same for the two samples taken in the Vacuum Degassing process.

Reply: Thanks for your valuable comments. The molten steel is hoisted to VD station after LF refining, and this process would take 5 to 10 minutes. The molten steel is violently moved and the slag-metal interface is strongly disturbed. Samples are taken to analyze the inclusions at the beginning of VD process. During the Vacuum Degassing process, the bottom blowing and stirring are continued, and the slag-metal reaction is intense, which would constantly change the composition and type of inclusions. Therefore, samples are taken to analyze the composition and type of inclusions after VD treatment, which are compared with that of beginning of VD process.

(5) Comment: Lines 71-77: Has a quantitative analysis of MgO-based inclusions been carried out at each stage of the steelmaking process? I think it would be interesting to define the stage in which the formation of a greater number of this type of inclusions occurs.

Reply: Thanks for your valuable comments. There are a certain number of MgO based inclusions in the whole refining process. The quantitative analysis of this type of inclusions at each stage of the steelmaking process has not been studied in this paper, and there are plans to study this process in detail in the future.

(6) Comment: Line 85. Table 3: I don't understand the information in the table. How is density calculated? What does the first column mean? Different samples analyzed? Does "Total" refer to the number of inclusions observed in each sample? Explain the information in the table more clearly, please.

Reply: Thanks for your valuable comments. The density is number of inclusion per square millimeter,and Table 3 has been corrected in the revised manuscript.

(7) Comment: Lines 86-88: Please describe in the text the characteristics (morphology and composition) of typical billet inclusions.

Reply: Thanks for your valuable comments. The corresponding description has been added in the revised manuscript.

(8) Comment: Lines 133-138: The formation mechanism of the MgO-based inclusions is consistent with the results of the thermodynamic analysis. If correct, I think that a lower number of MgO-based inclusions should be observed in the samples at the beginning of the Ladle Refining process. Do the authors have any evidence that this relationship?

Reply: Thanks for your valuable comments. The average MgO content of oxide inclusions increases from 7% to 15%. The average MgO content in inclusions increased rapidly from the late refining stage to VD stage.

(9) Comment: Line 153: “but the composition was relatively uniform”. What do the authors mean by this sentece? Explain it more clearly.

Reply: Thanks for your valuable comments. The composition of MgO was relatively uniform distribution.

(10) Comment: Line 157. Table 5: Same comments as those made in table 3.

Reply: Thanks for your valuable comments. Table 5 has been corrected in the revised manuscript.

(11) Comment: Lines 158-159: Please describe in the text the characteristics (morphology and composition) of typical billet inclusions. Compare the characteristics of the inclusions in Figure 9 and Figure 3.

Reply: Thanks for your valuable comments.  Before modification, it can be seen that the light gray part of the inclusion corresponds to the CaO-SiO2-Al2O3 phase and the black area depicts the MgO phase. After modification, the composition of Mg was relatively uniform distribution. The corresponding description has been added in the revised manuscript.

We have improved the manuscript in the manuscript. These changes will not influence the content and framework of the paper. And here we did not list the changes but marked in red in revised manuscript. We would like to express our great appreciation to you and reviewers for comments on our paper. Looking forward to hearing from you.

Thank you and best regards.

Sincerely yours,

Prof. Yanping Bao

State Key Laboratory of Advanced Metallurgy

University of Science and Technology Beijing

Beijing 100083, China

Jul. 8, 2022

Round 2

Reviewer 1 Report

THE MANUSCRIPT HAS BEEN IMPROVED AFTER REVISING.

Author Response

We have improved the manuscript in the manuscript. These changes will not influence the content and framework of the paper. And here we did not list the changes but marked in red in revised manuscript. We would like to express our great appreciation to you and reviewers for comments on our paper. Looking forward to hearing from you.

Thank you and best regards.

Sincerely yours,

Prof. Yanping Bao

State Key Laboratory of Advanced Metallurgy

University of Science and Technology Beijing

Beijing 100083, China

Jul, 25 2022

Reviewer 2 Report

REVIEW COMMENTS

Manuscript ID: metals-1783655_v2

Source and Transformation of MgO-Based Inclusions in Si-Mn Killed Steel with Lime-Silicate Slag

The manuscript has been significantly improved. I am grateful that the authors have taken into account the suggestions made by this reviewer. Grammatical errors have been corrected and the information is better structured. The scientific quality of the work has also been improved. This makes the manuscript easier to understand. New information has been added, such as Figure 2, which makes the results easier to understand. The authors describe in greater detail the results obtained and discuss them in a clearer and more rigorous manner.

Thank you for the work done.

Author Response

(The authors gave the same response as above.)
